# Neurorehabilitation of Spatial Memory Using Virtual Environments: A Systematic Review

**DOI:** 10.3390/jcm8101516

**Published:** 2019-09-20

**Authors:** Jessica Isbely Montana, Cosimo Tuena, Silvia Serino, Pietro Cipresso, Giuseppe Riva

**Affiliations:** 1Department of Human Sciences for Education, University of Milano-Bicocca, Piazza dell’Ateneo Nuovo 1, 20126 Milano, Italy; 2ATN-P Lab, Department of Psychology, IRCSS Auxologico Italiano, Via Magnasco 2, 20149 Milano, Italy; cosimotuena@gmail.com (C.T.); p.cipresso@auxologico.it (P.C.); giuseppe.riva@unicatt.it (G.R.); 3MySpace Lab, Department of Clinical Neurosciences, University Hospital Lausanne-CHUV, CH-1011 Lausanne, Switzerland; silvia.serino@chuv.ch; 4Department of Psychology, Catholic University of the Sacred Heart, Largo Gemelli,1, 20100 Milan, Italy

**Keywords:** navigation, neurorehabilitation, spatial memory, systematic review, virtual environment, virtual reality

## Abstract

In recent years, virtual reality (VR) technologies have become widely used in clinical settings because they offer impressive opportunities for neurorehabilitation of different cognitive deficits. Specifically, virtual environments (VEs) have ideal characteristics for navigational training aimed at rehabilitating spatial memory. A systematic search, following PRISMA guidelines, was carried out to explore the current scenario in neurorehabilitation of spatial memory using virtual reality. The literature on this topic was queried, 5048 papers were screened, and 16 studies were included, covering patients presenting different neuropsychological diseases. Our findings highlight the potential of the navigational task in virtual environments (VEs) for enhancing navigation and orientation abilities in patients with spatial memory disorders. The results are promising and suggest that VR training can facilitate neurorehabilitation, promoting brain plasticity processes. An overview of how VR-based training has been implemented is crucial for using these tools in clinical settings. Hence, in the current manuscript, we have critically debated the structure and the length of training protocols, as well as a different type of exploration through VR devices with different degrees of immersion. Furthermore, we analyzed and highlighted the crucial role played by the selection of the assessment tools.

## 1. Introduction

Virtual reality (VR) is a computer application by which humans interact with computer-generated environments in a way that simulates real life and involves various senses [1] and gives the user an experience of being “immersed” in the VR [2,3]. The experience created in VR depends on output tools (visual, aural, and haptic) that immerse the user in the virtual environments (VEs), input tools (trackers, gloves, or mice) that continually track the position and movements of the users, and the human interaction [4,5]. The degree of physical stimulation impacting on the sensory systems and the sensitivity of the system to motor inputs characterize the immersion experience. The product of immersion is *presence*, defined as the psychological sensation of “being there” in the VE instead of the physical and real environment [1,2,6] or as the “feeling of being in a world that exists outside the self” [4,7,8,9,10]. The most commonly used forms of sensory stimulation in VR systems are visual displays. A virtual camera controls the viewpoint from which the subject experiences the computer-generated image. The user’s perspective changes according to where he is looking; therefore, it is indispensable to track their location by an incorporated, highly sensitive head and body tracking systems. Sensors monitor the subject’s position to provide an egocentric reference frame for the simulation. The images can be delivered either by a head-mounted display (HMD) or by a computer monitor or projection screen. HMDs may be more immersive but can induce cybersickness in vulnerable subjects, whose symptoms are a headache, eye strain, nausea, or, in extreme cases, vomiting [11,12]. In this sense, display screens, semi-immersive systems, are generally more comfortable to use. Auditory and haptic stimulations are often combined with a visual display and are increasingly able to provide a strong sense of physical contact with the VE [13]. In less immersive systems, the input is retrieved from standard joystick controllers, mouse, and keyboard. These control devices are easy to use and naturalistic interfaces that simulate real-world interactions are largely used [1]. The use of VR in neurorehabilitation has grown in a meaningful way, and experimental evidence suggests that this technology could have a positive impact on functional recovery in neuropsychological disorders [8,14]. It is a fascinating tool in neurorehabilitation for its peculiarities. First, the possibility of creating tailor-made training that has the value of highlighting how each rehabilitation process must be individualized, addressing the recovery of the patient’s specific disorder and adaptation request [1]. The active involvement that this tool can generate in the subjects is due to the possibility of creating new and appealing environments without forgetting the valuable immediate and concrete feedback that comes to the person. The movement accomplished can be reproduced by the avatar within the VR and this is crucial feedback for the patient [15,16,17,18]. VR also offers the opportunity for controlled, ecological, and secure testing environments with different degrees of immersion and interaction [1,19]. Thus, an increasing number of cognitive rehabilitation programs have started using VE to simulate daily activities, such as shopping, traveling [20], or exploring a city [21].

The literature shows that virtual reality is an acceptable and promising therapeutic tool for several pathological fields [13], such as mental health disorders in patients with post-traumatic stress disorder, anxiety and depression [22,23], or eating disorders [14,24,25] and in neuropsychological deficits, for instance, in patients with traumatic brain injury (TBI) [26]. Interesting are the results in which VR has been shown to have potential for improving the assessment and treatment of TBI and dementia [27,28,29], even in cases where the probabilities of recovery appear low [30]. It has been demonstrated as a successful tool in spatial memory and navigational abilities, particularly in Alzheimer’s disease (AD) and in mild cognitive impairment (MCI) [13,31,32].

Spatial memory is reflected in a person’s navigation and orientation abilities, fundamental requirements for moving in the surrounding environment. The ability to reach landmarks efficiently depends upon the ability to form, retain, and utilize a cognitive representation of the environment [33]. Human navigation involves several cognitive functions and processes. It can be based on self-motion cues and static environmental cues. The tracking of a person’s position and orientation is based on self-motion cues, motor efferences, and vestibular and proprioceptive feedback [34]. Environmental cues are based on landmarks and extended boundaries that can provide one’s position and orientation relative to the environment. Self-motion and static environmental cues can inform allocentric and egocentric reference frames [35,36]. Allocentric representation is independent of the position of the navigator and does not change with the navigator through space. An egocentric frame, however, involves the representation of locations based on the subject’s viewpoint [37]. The self-reference system uses self-motion cues to update body location and face direction relative to an allocentric, orientation-free, immediately available, object-to-object map [38].

Spatial memory problems, such as forgetting the orientation and the position of objects or getting lost, are often a result of hippocampal damage in humans [32,37,39,40]. The consequence of these representations can be dissociated in terms of behavioral and developmental elements, and, finally, of their neural bases. Thus, the hippocampus and medial temporal lobe offer allocentric environmental representations, whereas the parietal lobe egocentric representations and the retrosplenial cortex and parieto-occipital sulcus allow both types of representation to interact with each other [37,39,41,42]. In human navigation, the role of the hippocampus and associated mesial temporal lobe structures has been widely demonstrated [37]. Meanwhile, differential activity in the hippocampus and caudate correspond to the acquisition and expression of information about locations derived from environmental boundaries or landmarks, respectively [43]. Changes in the navigation network may be a result of cognitive decline and can manifest in impaired spatial navigation [35].

In conclusion, the environments recreated using VR technology represent a context through which the user has the opportunity to experience real-life scenarios and increase their abilities and experience new adaptation strategies [13]. In [44], it is suggested that patients are able to transfer information about the environment acquired from VE to real life. They suggest that mental representations of space in VE are rather like those implicated in the navigation of the real world. Concerning the growing interest in VR and high potential applications in neurorehabilitation, it is necessary to examine the treatment procedures and the results obtained so far. According to these premises, we aimed at providing a systematic review of the experiments in the field of spatial memory neurorehabilitation to comprehend if VR navigational training, compared to treatment as usual, is effective in improving navigational abilities. The specific objectives of the present work are two-fold. First, to provide an overview of which apparatus are available for neurorehabilitation and understanding how these VR training regimes have been implemented in clinical settings. We analyzed different types of software and procedures for implementing the training. Finally, in light of the cognitive and neural theories of spatial processing, we attempted to compare different VR navigational training used and analyzed which is more useful.

## 2. Method

### 2.1. Search Methodology

Preferred Reporting Items for Systematic Reviews and Meta-Analysis (PRISMA) guidelines were followed [45]. Two high-profile databases (PubMed and Web of Science) were used to perform the computer-based research on the 6 July 2019. According to PICO format, we defined the question (following the identification of problem, intervention, comparison group, and outcome) as “In spatial memory disorder, is VR navigational training, compared to treatment as usual, effective in improving navigation abilities?”. We then proceeded with the definition of keywords for the search strategy. The string used to carry out the search strategy was virtual-realit* OR virtual-environment* AND neurorehabilitation OR rehabilitation OR training OR stimulation OR navigation OR learning OR abilit* OR memor* AND spatial OR space. 

From the search of both databases, we obtained 5048 articles, excluding duplicates. Title and abstract screening was carried out, and 24 articles passed to the full-text screening phase. Eight studies were excluded with reasons as follow: Not Controlled trial (= 1); Results of neuropsychological outcome for spatial memory not reported (= 3); Qualitative/descriptive study (= 2); Not neuropsychological rehabilitation for spatial memory (= 2).

### 2.2. Inclusion Criteria

The review considered randomized control trials, nonrandomized control trials, intervention studies, and case-control studies in clinical patient populations with an overt spatial memory disorder. Studies on rehabilitation’s programs of navigation abilities with virtual reality (VR) devices in different population of patients (such as mild cognitive impairment (MCI), Alzheimer’s disease (AD), traumatic brain injury (TBI), multiple sclerosis (MS), stroke, cerebral palsy, epilepsy, incomplete cervical vertebro-spinal trauma, topographical disorientation disorders and neglect) were included. We only included studies in the English language and which satisfied strict criteria for eligibility for the review (research studies, clinical patient population, VR training, spatial memory disorders, rehabilitation programs). The qualitative component also considered the type of VR navigational training and methodological design. We excluded articles which lacked necessary information for review in the full-text or the abstract. Reviews, meeting abstracts, proceedings, notes, case reports, letters to the editor, assessment protocols, editorials, and other editorial materials were also excluded. Retrospective studies were not included because the area of interest requires performing experiments. 

### 2.3. Risk of Bias Assessment 

To assess the risk of bias, the reviewers followed the methods recommended by The Cochrane Collaboration Risk of Bias Tool [46] and the STROBE Statement [47]. Two reviewers (J.M. and C.T.) independently assessed the risk of bias of each included study against key criteria: random sequence generation, allocation concealment, blinding of participants, personnel, and outcomes, incomplete outcome data, selective outcome reporting, and other sources of bias. The following judgments were used: low risk, high risk, or unclear (either lack of information or uncertainty over the potential for bias). Disagreements were resolved through consensus, and a third author was consulted to resolve disagreements if necessary. In particular, the selected studies followed strict criteria in the methods, including presenting critical elements of study design, clearly defining all outcomes, describing the setting and relevant dates, including periods of recruitment and exposure, giving sources of data and details of methods of assessment (measurement).

### 2.4. PRISMA Flow Diagram

PRISMA guidelines were strictly followed; all titles and abstracts were screened according to the abovementioned inclusion criteria after removing the duplicates. Full texts of eligible articles were retrieved and assessed by two reviewers (J.M. and C.T.) for individual selection of papers to reduce the risk of bias and resolving disagreements through consensus as explained in Section 2.1. See Figure 1 for the paper selection procedure. 

## 3. Results

Sixteen studies were analyzed to test the usefulness of rehabilitative interventions using virtual reality (VR) systems. However, growing interest in VR has led researchers to question how the characteristics of VR equipment and different aspects of the training tasks could influence the treatment outcomes, with particular regard to the results that reflect on the patient’s daily life in an ecological way. In our review, we aim at giving more awareness to the researchers and at guiding them in the selection of the most appropriate VR device to use. Considering the studies mentioned in this analysis, it could be possible to understand which is the most suitable program for the treatment of spatial memory disorders, in terms of the type of apparatus used and the training method. Furthermore, it is essential to understand which kind of patient will benefit from the intervention. To satisfy our aims and to facilitate the understanding, we considered the following clusters: (1) Authors; (2) Year; (3) Sample (N); (4) Sample characteristics; (5) Mean age; (6) VR Task; (7) Virtual Apparatus; (8) Neuropsychological assessment; (9) Primary Outcomes. Results are reported in Table 1.

Analysis of results in spatial navigation rehabilitation programs was carried out starting from the outcomes of 16 studies, taking into consideration the patient population, the VR apparatus (immersive/semi- or nonimmersive) and the type of training used. We analyzed clinical, methodological, and technical outcomes to cast research and clinical applications of VR in the context of VR navigational training. The results were argued in response to the following three questions. 

### 3.1. Which Virtual Apparatus Is Recommended for Spatial Memory Rehabilitation?

#### 3.1.1. Type of Device and Controllers During Navigation

Among all the studies reviewed, the only one that used an immersive virtual reality (VR) system with a head-mounted display HMD Oculus Rift DK2, using the Unity 5 game engine, and a joypad is described in [39] (see Table 1 for an overview of VR equipment). Several studies [20,52,57,58,59,61], used semi-immersive systems: the previous one used OctaVis, a circle of eight screens in which the participant can freely rotate on a fixed chair and interact with a joystick; while [58,59] used a BTs Nirvana PC System connected to a projector or a big screen and to an infrared sensor for movements; in [61], a finger touch projector was used and the scenarios were developed using Unity 3D. In order to increase the user’s presence in the VE, attention was paid to elements such as the visual flow synchronized with the cycling velocity in real-time, realistic 3D sound, perception of the wind through the movement of trees; [57] used a projector on a big screen (2 × 1.5 m) and a viewing height of 1.8 m. Virtual Scene Designer was used to create the scenarios. The advantage conferred by the semi-immersive system is that it allows sensorial isolation, but it presents fewer signs of cybersickness. Even this system demonstrates the generalizability of VR measures by correlations with subjective estimations of cognitive abilities and real-life shopping performance [52]. Immersive and semi-immersive systems have proved to be useful tools in navigational training for improving spatial cognition and attention processes [55,58].

Nine further studies reviewed used nonimmersive systems. In [50] the rehabilitation training was based on a navigational task, exploring part of a virtual town (London) from a ground-level perspective, using a computer videogame driving simulator (Midtown Madness 2, Microsoft Game Studios). Other studies have used nonimmersive VR training on a computer with joystick or keypad to navigate the city, using Unity 3D software—Reh@city [54,56]—on a standard IBM-PC computer and in an environment developed with the Super Scape VRT-3D construction package [49]; or a specific software developed by authors—Virtual Tübingen [53]. Other studies used Superscape software version 4 and NeuroVR. NeuroVR is a free VR-platform for customizing a large number of predeveloped virtual environments [9,21,48,62]. One study [51] created a simulation of the real-world town of Graz (Austria) using Instantreality software, a high-performance Mixed-Reality framework that provides a comprehensive set of features to support classic virtual reality [63]. Another study [60] used a certified medical device with high customization capacity, the virtual reality rehabilitation system (VRSS), which comprises a central hub connected to specialized peripheral devices, such as magnetic sensors for movements, which is fully synchronized and integrated with the system. Also, the rehabilitation programs that used a nonimmersive system showed promising enhancement of different cognitive abilities through spatial and navigational training [21,48,50,51,52,53,58].

#### 3.1.2. VR Spatial Navigation

The usefulness of VR navigational tasks in rehabilitation programs for spatial memory has been highlighted. Promising results have been demonstrated with the use of navigation tasks on simple display screens of a computer (nonimmersive devices). However, better performance can be achieved through immersive rather than semi-immersive systems by using head-mounted displays. This is explained by the greater sense of presence [4,8], with integrated systems that record the user’s movement and take advantage of up to six degrees of freedom, and providing feedback to the user about their performance [15]. Since navigation in virtual environments can trigger the same brain mechanisms as navigation in the real world, spatial “presence” can be generated [64]. 

### 3.2. Which Virtual Training Method Is Suitable for Spatial Memory Rehabilitation?

The studies included in the current review are all focused on the rehabilitation of spatial memory and navigation abilities in different neurologic patients, including mild cognitive impairment (MCI), Alzheimer’s disease (AD), traumatic brain injury (TBI), multiple sclerosis (MS), stroke, cerebral palsy, epilepsy, incomplete cervical vertebro-spinal trauma, prosopagnosia with topographical disorientation disorders, and neglect. all patients have used virtual reality (VR) devices in different rehabilitation programs, and nobody had any difficulty in following the training. Overall, the emerging outcomes are positive, and they present improvements to different degrees. All of the mentioned studies had a control group [20,21,48,49,51,52,54,57,59,61] except one [53]. Five of them reported results of single case studies [50,55,56,58,60]. Virtual rehabilitation enables clinicians to control the specific features of the virtual environment, enabling tailoring of the challenge to suit individual patient needs [65]. The characteristics of virtual training (detailed in the following paragraphs) that are crucially important are the overall duration of the training, the frequency, the intensity of each session, and, last but not least, the time elapsed since the damage [57]. Additionally, some studies have already demonstrated that the use of a map as a navigational aid improved the performance of users performing complex navigational tasks [66]. Furthermore, the presence of a small-scale interactive aerial view facilitated the retrieval of stored spatial layout and an arrow or salient landmarks, giving more comprehensive information about the egocentric heading in environment, were effective in supporting the navigation [67,68]. Also, findings in the studies examined underline the importance of using active navigation protocols to promote the neurorehabilitation of spatial memory [62,67], and that the degree of the visual similarities between the virtual world and the real one boosts the transfer of learning between contexts [57].

#### 3.2.1. VR Training Duration

The protocol duration turned out to be an important variable for rehabilitation training outcomes (details are visible in Table 2). The training proposed by [50,60] was the most intensive. The first regime consisted of 15 sessions of 90 min each, amounting to 22.5 h. The second regime consisted of three weekly sessions of 60 min for a total of 36 treatments. The improvements are visible after a certain period of time, as evidenced by two-month and one-year follow-ups [50]. One study [55] proposed 21 sessions of 45 min each, amounting to 15.75 h, and [61] proposed 18 sessions lasting 40–45 min each for a total of 13.5 h. In these studies, the patients showed substantial improvement in navigation ability. Another study [59] conducted 20 sessions for stroke patients in which an intensive and long training program was essential for obtaining substantial improvements. Other rehabilitation programs, which lasted between eight to 15 sessions, showed an improvement in long-term spatial memory after VR-based training [21,52,57]. Transference of improvements from the VR-based training to more general aspects of spatial cognition was observed [21]. Interesting results are also connected with high frequency and intensity of sessions [21,50,52,55,58,59] rather than low-intensity training regimes [49]. A protocol lasting less than four hours returns vulnerable outcomes [20,51,53,54,56,62]. For example, one study [53] based on four sessions of 1 h each was only able to significantly improve one patient’s performance. 

#### 3.2.2. Time Elapsed Since Damage

Another important variable is the time elapsed from the brain injury to the starting point of cognitive rehabilitation. One study [54] found an overall recovery and showed a positive training effect on global cognitive functions in post-stroke patients which started the treatment within 7 months from the stroke. A long-distance from the traumatic event, instead, does not promise good results, as happened in patients who started an average of 43 months after stroke [53]. A short interval was also important in other disorders. One study [50] have gained promising results with a single traumatic brain injury (TBI) patient that began the training within 1 year. The shorter the period between the traumatic event and the beginning of treatment, the higher the probability of achieving a better outcome [50,52,54].

#### 3.2.3. Training Procedure

A detailed description of the different procedures is outlined below. In one study [50], the patient was given the instruction “You must cut down poles and trees that you find all along the way”, with the aim of inducing the participant to explore the virtual town, avoiding passing the same roads twice. The spatial memory task was implicit, while the explicit task was a simple game, which entertained the participant. The participant did not have access to a city map during the task.

One study [53] used training divided into implicit (free exploration) and explicit tasks (following specific routes). During the sessions, the patients were encouraged to practice the instructed navigational strategy learned in a previous psycho-educational phase. The participants had access to a city map during the task. Two studies [54,56] used implicit tasks. In the first study, the participant was guided by instructions such as “Go to the supermarket”. Visual feedback elements (time and point counters) were used to reward successful actions. The participant had access to a mini-map in the lower half of the screen and/or a guidance arrow. In the second study, the participant was asked to follow three routes, differing in length, as quickly and as accurately as possible. In the next learning phase, the patient was required to create a mental representation of the city, incorporating the spatial location of six landmarks. In the retrieval phase, he was required to navigate via the city in order to reach a location as quickly as possible and using the shortest route possible. 

In [55], the participant was instructed to enter a building and find the correct window, which had been previously shown from an external view of the building. Two studies [21,51] used specific tasks with encoding and retrieval phases. In the first study, the neuropsychologist asked participants to find and memorize the position of hidden objects within the virtual city. Then, they were asked to retrieve the position of the objects identified before, starting from another point of the city. Participants had access to a city map during the task. In the second study, the participant had to learn a route by following verbal instructions and subsequently had to recall the correct directions. In two studies [48,57], the participants received explicit instructions to explore and memorize a route in the virtual environment (VE) during the research of an object. Subsequently, in the first study, they were asked to draw the layout of the VE, and in the second study they had to move through the equivalent real-life environment. Also, in [49], the participants were asked to move through a maze to reach a tree. In the first and second versions, the plan of the maze was visible, and the target always remained visible. In the third version, the target tree could be seen only from a short distance when not hidden by the maze walls. In [20,52], the participants had to memorize an auditory shopping list of 20 items as an implicit spatial memory task and subsequently buy all the items remembered in a VR supermarket. In the second study, the same authors presented a new interfering list with 20 distractor items during the training.

Three studies [58,59,60] included a series of exercises involving different cognitive functions. The participants were asked to remember the positions and the name of elements observed or to program movements to manipulate specific objects and to realize specific associations with dynamic interaction in VE. In [61], three scenarios were used, in which patients had to navigate freely to accomplish the task explicitly requested, for example, purchasing five items form a supermarket. The spatial memory task was explicit, and only in the last version was no aid was given for completing the task. In conclusion, independently from the simulation, the outcomes showed that VR training enhanced spatial memory abilities in the clinical population. 

#### 3.2.4. Visual Cues

In terms of guide elements which facilitate the patient during the training, different cues have been included in some of the tasks, such as maps [21,53,54], guidance arrows [54], and lists of objects. In particular, [54] employed a method of fading cues, decreasing assistance (DA), in which the training continues with all the cues until correct performance is achieved on three consecutive sessions and then they are gradually removed. Furthermore, in [21], the number of objects to be memorized depended on the level reached by each participant; if the patient was not able to locate the first object, the other ones were not presented. The task presented to the participant was guided in some cases by explicit navigational instructions [21,53,56] that needed to be followed to memorize the route.

### 3.3. Which Assessment Method Is Best for Spatial Memory?

#### 3.3.1. Spatial Memory Outcomes

In clinical practice, the most common neuropsychological test for the evaluation of short and long-term spatial memory, given its psychometric characteristics, is the Supraspan Corsi Test. In these studies, the Supraspan Corsi Test has proved to be adequately sensitive and has indicated a significant improvement in spatial memory [50], with a medium effect size *r* = 0.474 and *p* = 0.03 [21]. Another ecological spatial memory assessment tests, Route delayed recall (RBMT), has generated desirable changes (0/5 to 4/5) that persisted at 2-month and 1-year follow-ups (respectively 4/5 and 5/5) [50]. One study [53] has shown that the virtual reality (VR) system is a sensitive assessment tool for the same cognitive function (Virtual Tübingen Test). They found that one patient improved in nine out of the 10 virtual navigation subtasks. To assess spatial navigation and memory performance in the VR supermarket, [52] analyzed the enhancements in “number of correctly bought products”, “number of the correct products relative to distance”, and “number of the correct products relative to time”. Other authors asked the participant to form a cognitive map of the VE, and active participants showed significantly better performance [48] and were quicker [56] than the control group. A similar performance pattern was observed at the one-week follow-up session. VR neurorehabilitation programs for spatial memory can also provide a positive effect on other cognitive domains. By using specific-domain assessments, it is possible to observe if a transference of improvements occurs from VR-based training to more general aspects of spatial cognition. Following the treatment, however, general enhancement of cognitive functions occurs and is reflected in several assessment tests. The Rey Auditory Verbal Learning Test (RAVLT) has shown significant improvement in the immediate recall that persists at 1-year follow-up [50]. A considerable improvement can be observed in screening tests, such as in Addenbrooke’s Cognitive Examination (ACE) (*r* = 0.85, *p* = 0.011), particularly in visuospatial (*r* = 0.80, *p* = 0.017), attention (*r* = 0.79, *p* = 0.018), and memory (*r* = 0.79, *p* = 0.017) domains and in the Mini-Mental State Examination (MMSE) (*r* = 0.75, *p* = 0.025) [54]. Montreal Cognitive Assessment (MoCA) results significantly improved in the experimental group after treatment in visuospatial and attention domains [59]. However, [55] did not observe consistent changes in MoCA. However, after the training, the participant was able to complete the VRN Building navigational task correctly, and the effects persisted at 5- and 28-week follow-ups. Improvements in navigational abilities have also been confirmed in daily life as recorded in wives’ diaries. In [51], the patient group showed significantly higher scores in the Achievement Measure System LSP 50+, a German standardized intelligence test developed for older people between 50 and 90 years, and in the Benton Visual Retention Test, which assesses visual perception and visual memory. Also, in the Visual Pursuit Test (LVT) for visual orientation assessment, the authors observed a significant result [51]. De La-Torre [57] showed the existence of a main effect between the targets and the average errors committed to locate them within the virtual building (F adjusted (1.37, 28.01) = 8.55, *p* < 0.01; ŋ² partial = 0.32, observed potency = 0.87). One study [52] to analyze changes following treatment made correlations between VR measures and cognitive performances in the patient group. They observed that the mean number of correct products per time across all learning trials was correlated with higher performance on the Bergen Right-Left Discrimination Test (BRLD-B) to assess mental rotation. The percentile of the Digit Span Forward for short-term verbal memory was significantly correlated with the number of correctly bought products in the last learning trial on day 6 and number of correct products on the free recall trial after interference on day 7. The mean number of correct products per time and per distance across all learning trials were correlated with better performance on, respectively, the delayed free recall and immediate and delayed free recall of the Rey-Osterrieth Complex Figure (ROCF) for the assessment of visuo-construction, planning, and long term visual memory. A significant improvement was also observed in ROF [48]. In [56], the patients’ face recognition performance significantly improved as measure by the Cambridge Face Memory Test (CFMT) following training. The effects persisted at one-week and 6-month follow-ups.

#### 3.3.2. Traditional and Virtual Assessment

Results of the current review support the idea that virtual reality-based training improves orientation and navigation abilities in different neuropsychological disorders. Although the efficacy of treatment needs to be critically analyzed using a scientifically validated method, for this reason, the use of validated measures with adequate psychometric properties is fundamental. Over the past decade, evidence-based clinical guidelines have become a significant feature of healthcare services to improve the quality of care [69]. Indeed, among the studies analyzed, those that used specific-domain assessments (Corsi Supraspan, Virtual Tübingen, Route delayed recall RBMT, cognitive maps, or variables extracted from the supermarket task) were more able to demonstrate clear and significant results after the spatial memory rehabilitation training [21,48,50,53,56], highlighting the crucial role played by the selection of the assessment tools. In addition to the traditional paper-and-pencil tests used, new virtual assessment measures also emerged in the analyzed studies. These last have shown that the virtual reality (VR) system could be a sensitive assessment tool for detecting spatial memory improvements. Therefore, it is urgent to find more scientific evidence regarding the psychometric validity of these new measuring instruments, particularly concerning navigation abilities.

## 4. Discussion

In the current review, we provided initial, positive results concerning the effect of virtual reality (VR) training on spatial memory rehabilitation, highlighting the potential of navigational tasks in virtual environments (VEs) to enhance navigation and orientation abilities in patients with spatial memory disorders. The rapid development and diffusion of VR technologies are amending the accessibility landscape of VR technology for the average consumer. Lower-cost VR systems such as the Oculus Rift, Go, Quest, and the HTC Vive are already issued on the market and have significantly reduced the cost barrier of VR hardware. Even lower-cost options are currently available using a smartphone, for example, Gear VR is compatible with specific Samsung phones, and both Google Cardboard and Google Daydream can be used with several smartphones. Although the review underlined encouraging results, current research in this topic has some limitations that researchers need to overcome. The current work is meant to provide methodological solutions for future studies. 

As the first result of the review, we have found clear improvements in spatial memory through the application of navigational tasks in VR. Both immersive and nonimmersive VR systems have shown appropriate enhancements for navigation and orientation abilities, underling the power of the navigational tasks proposed. Furthermore, the results have shown a transference of improvements from VR-based training to more general aspects of spatial cognition. However, the mode of exploration influences the spatial learning of a new environment. The active exploration has an essential role in the acquisition of spatial knowledge and it is characterized by five components: motor orders that determine the path of locomotion, proprioceptive and vestibular information for self-motion, allocation of attention to navigation-related features of the environment, and cognitive decisions about the direction during navigation and mental manipulation of spatial information [70]. If immersive systems are able to target all these components, nonimmersive systems do not allow the activation of the idiothetic and motor systems, even if input devices require motor planning and execution. In this review, only one study [55] used an immersive virtual reality system with the Oculus Rift DK2 head-mounted display and a joypad. Instead, other studies [20,52,58,59] used semi-immersive systems: the first one used OctaVis, a circle of eight screens in which the participants can freely rotate on a fixed chair and interact with a joystick; the latter used the BTs Nirvana PC system connected to a projector or a big screen and an infrared sensor for movements. The recent availability of lower-cost options for immersive VR may change the situation soon, allowing more protocols that fully support active exploration.

Another important element for the rehabilitation program is the structure and length of the training protocol. Some studies have already demonstrated that the use of visual cueing, such as a map as a navigational aid, improved the performance of users performing complex navigational tasks [66] and in patients with Alzheimer’s disease or mild cognitive impairment [70]. The presence of an interactive aerial view facilitated the retrieval of stored spatial layout and arrows or salient landmarks, giving more comprehensive information about the egocentric heading in the environment, were effective in supporting the navigation [67,68]. Also, the current review underlined the importance of cues in the tasks, such as maps, guidance arrows, methods of fading cues, and instructions (explicit or implicit), to support the patient during the training [21,53,54]. It has been demonstrated that a large visual arrow supports the neurorehabilitation of spatial memory due to the cognitive synchronization between the allocentric viewpoint-independent representation (including object-to-object information) and the allocentric viewpoint-dependent representation (i.e., comprising information about the current egocentric heading in the environment), as suggested by the “mental frame syncing hypothesis” [32,71]. Furthermore, the duration of the protocol [50,60] and time elapsed since damage onset are also key factors [50,52,54]. The analysis of these studies showed that a short duration—less than four hours—is insufficient to provide consistent changes. Concerning the distance from onset, the shorter the period between the traumatic event and the beginning of treatment, the higher the probability of better outcomes.

The rehabilitation programs analyzed all focused on spatial memory and navigation abilities in different neurologic patients, including mild cognitive impairment (MCI), Alzheimer’s disease (AD), traumatic brain injury (TBI), stroke, epilepsy, prosopagnosia with topographical disorientation disorders, and neglect. An important aspect to keep in mind is the clear characterization of the sample of patients to whom the therapeutic procedure is provided. The severity and the qualitative characteristics of the mnestic deficit are extremely variable from subject to subject. Furthermore, memory disorders rarely occur in an isolated form, and are often accompanied by an impairment of other cognitive functions, such as attention, language, reasoning. In this view, the most effective protocols are the ones that target a specific pathology. 

The difficulty in finding homogeneous groups of patients is the reason why most of the experimental studies reported in the literature are based on the treatment of a single patient or a tiny group of patients. However, to monitor the improvements due to the treatment, the presence of an adequate control group is necessary to ensure that the observed improvement is not due to spontaneous recovery nor is the result of generic cognitive stimulation. In this view, the most reliable control condition is the repeated evaluation of the same patients receiving the experimental treatment, for example, through cross-over trials [72]. Another possible approach is the involvement of patients who, in the immediately preceding period, were followed in the absence of therapy or using another type of cognitive treatment. Moreover, the studies that used domain-specific assessments (Corsi Supraspan, Virtual Tübingen, Route delayed recall RBMT, cognitive maps, or variables extracted from the supermarket) were more able to demonstrate clear and significant results after the spatial memory rehabilitation training [21,48,50,53,56], highlighting the crucial role played by the selection of the assessment tools. Among the tests used (Corsi Supraspan, Virtual Tübingen, Route delayed recall RBMT), only the Virtual Tübingen can investigate the several aspects (scene recognition, route continuation/sequence/order/progression/distance, pointing to start/to end, map drawing recognition) involved in navigation. Thus, the scarcity of tools for spatial memory assessment is evident, particularly concerning navigation abilities.

This review was subject to certain limitations. Extensive literature research was performed to deliver these findings; however, only two databased were queried. A second limitation of this review may be the selection of the keywords in the search strategy. In order to provide a considerable quantity of studies in the field, the authors decided to use wider keywords. However, the inclusion criteria reduced the focus to only studies according to the objectives of the review. A final limitation may be the fact that the discussion on duration and intensity of VR navigational training cannot draw firm conclusions due to the heterogeneity of interventions. Moreover, the difficulty of finding a homogeneous group of patients is the reason why most of the experimental studies reported in the literature are based on the treatment of a single patient or a tiny group of patients. Because our findings are positive, we hope that the literature could soon be enriched with clinical trials investigating VR’s effectiveness in more specific and larger clinical populations.

## 5. Conclusions

In conclusion, the results of this systematic review demonstrate that all studies, although to varying degrees, suggest that patients improved their spatial memory following treatment. This result highlights the potential of navigational tasks performed in virtual environments (VEs) for enhancing navigation and orientation abilities in patients with spatial memory disorders. In neuroscience, researchers have long faced the challenge of conducting ecologically valid measurements of experimental variables while maintaining strict experimental control over visual displays. Virtual reality (VR) systems enable the researchers to design and consequently control dynamic, realistic, and immersive environments, while closely monitoring behavioral and physiological responses during experimentation [1,73]. In this view, VR systems offer impressive opportunities as an ecological tool which is currently available for neuropsychologists to assess and enhance spatial memory, particularly navigation and orientation abilities using the “affordances” [74] offered to the patients in the virtual environment. One of the significant advantages of VR is the high degree of experimental control that is afforded to investigate the cognitive and behavioral components that are involved in spatial navigation [75]. VR training can facilitate neurorehabilitation, promoting brain plasticity processes through complex mechanisms related to the reactivation of brain neurotransmitter capacities, maximizing the results compared to those obtained by conventional treatment [66,76]. Monitoring EEG activities of the patients could be a suitable practice to assess the biomarkers of neuroplasticity and to measure rehabilitation progress [67,77]. The results are promising; hence, we encourage researchers to develop new spatial memory VR-based protocols for neurorehabilitation. However, more research is required to validate and support this result.

## Figures and Tables

**Figure 1 jcm-08-01516-f001:**
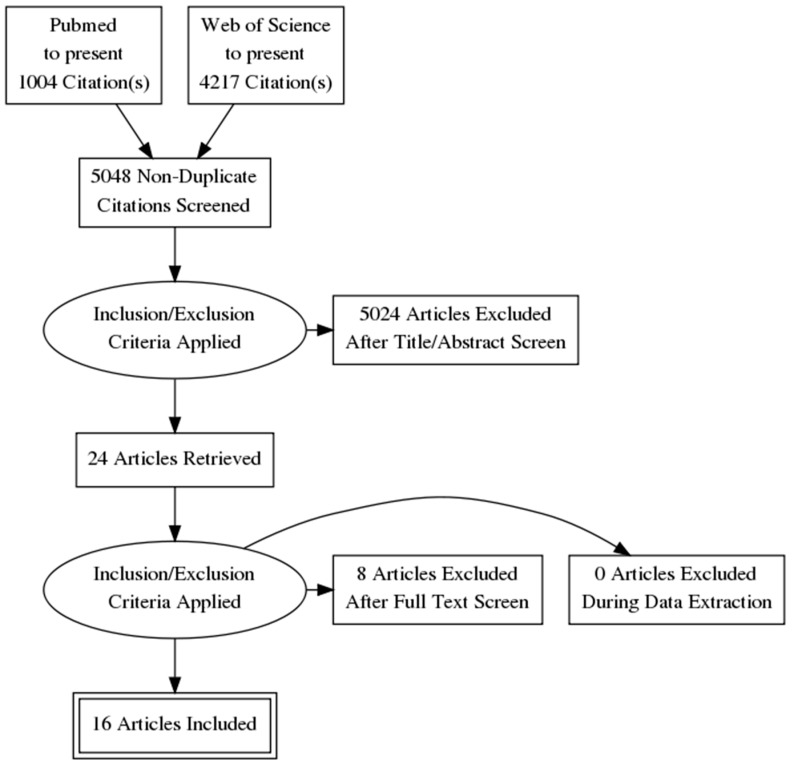
PRISMA flow diagram.

**Table 1 jcm-08-01516-t001:** Papers included in the PRISMA systematic review. MMSE—Mini-Mental State Assessment; MoCA—Montreal Cognitive Assessment; ACE—Addenbrooke’s Cognitive Examination; ADAS—Alzheimer’s Disease Assessment Scale; WAIS-III—Weschler Adult Intelligent Scale; RAVLT—Rey Auditory verbal Learning Test; RBMT—Rivermead Behavioural Memory Test; CBTT—Corsi Block-Tapping Test; DART—Dutch Version of Reading Test; BIT—Behavioural Inattention Test; FIM—Functional Independence Measure; FAB—Frontal Assessment Battery; AM—attentive Matrices; TMT A- B- A/B—Trial Making Test; BIT—Behavioural Inattention Test; WMS IV—Wechsler memory scale; WTAR—Wechsler Scale of adult Reading; VOSP—Visual Object and Space Perception Battery; CFMT—Cambridge Face Memory Test; CFPT—Cambridge Face Perception Test; BRLD—Bergen Left-Right Discrimination Test; ROCF—Rey–Osterrieth Complex Figure Test; RWT—Regensburg Verbal Fluency Test; VLMT—German adaptation of the Rey Verbal Learning Test; LPS 50+—The Achievement Measure System 50+; LVT—Visual Pursuit Test.

	Authors	Year	Sample (N)	Sample Characteristics	Mean Age (SD or Range)	VR Task	VR Apparatus	Pre- and Post- Assessment	Primary Outcomes
1	Pugnetti et al. [48]	1998	30	**Experimental Group (EG)** 15 MS patients. **Control Group (CG)** 15 healthy controls.	**EG** active condition mean age = 39.1; Standard Deviation (SD) =11.1/passive condition mean age = 37.7; SD = 8.1. **CG** active condition mean age = 35.8; SD = 9.41/passive condition mean age = 35.4; SD = 12.2	The aim was to explore the VE of a house, composed of four rooms and corridors, in search of an object.	Nonimmersive Virtual Reality (Superscape Software, version 4).	ROF, CBTT, Raven’s matrices IQ.	Spatial memory improved in the active subject (MS and healthy) suggesting that direct interaction with the environment can enhance navigation ability.
2	Akhutina et al. [49]	2003	EXP 1. 21 EXP 2. 45	EXP 1. **EG**/**CG** 21 patients with a diagnosis of cerebral palsy. EXP 2. **EG**/**CG** 45 patients with a diagnosis of cerebral palsy.	EXP 1. **EG** 12 (range 7–14) **CG** 9 (range 7–14) EXP 2. **EG** 23 (range 7–14) **CG** 22 (range 7–14)	The aim in each version of the task (drawn, real or virtual) was to move through a maze to reach a tree.	Non immersive environments IBM-PC and a mouse (Super Scape VRT 3-D Software) displayed on a 40,630 cm monitor.	EXP1. computer versions of the Koos Block Design Test, and a Clown Assembly Test. Decentration of Viewpoint Test, and Directional Pointing to a Hidden Object Test. EXP 2. Additional measures: Raven Progressive Matrices; The Benton Judgment of Line Orientation Test; The arrows subtest of the Nepsy; The Roads Test.	The studies have demonstrated that the general spatial abilities of a group of children with motor disabilities can be enhanced using a battery of training tasks that demand the use of various spatial skills. The battery included VEs that provided the children with navigational spatial experience, of a kind that most would rarely (if ever) experience in the course of their daily lives.
3	Caglio et al. [50]	2012	1	TBI patient with hemorrhagic contusions in the bilateral frontal, temporal and parietal lobes.	24 (male)	The aim was to explore part of a virtual town (London) from a ground-level perspective.	Nonimmersive Virtual Reality (Midtown Madness 2 videogame).	Corsi Block-Tapping Test, Corsi Supra-Span Test, Backward digit span, RAVLT, TMT A-B, Phonemic fluency, ADAS, RBMT.	Improvement in immediate verbal learning, immediate and delayed spatial learning and everyday-spatial memory persisted at follow-ups.
4	Grewe et al. [20]	2013	24	**EG** 5 patients with focal epilepsy (2 right temporo-parietal; 1 right hippocampal; 1 bilateral temporal; 1 bilateral occipital periventricular). **CG** 19 healthy participants	**EG** mean age = 35.04; SD = 8.08; **CG** mean age = 23; SD = 3.45	The aim was to navigate into a virtual medium-sized supermarket, modeled according to a real standard supermarket, in search of a specific list of objects.	OctaVis, semi-immersive Virtual Reality device.	ROF	The supermarket training provided preliminary evidence of effectiveness, but significant improvement was not found. A strong limitation was due to the small sample size.
5	Kober et al. [51]	2013	23	**EG** 23 patients: 3 right and 1 arteria cerebri media stroke, 1 basal ganglia and thalamus stroke, 1 right arteria cerebri media, 1 basal ganglia stroke, 1 right fronto-parietal stroke, 2 right aneurysm and subsequent infarct (arteria cerebri posterior and arteria communicans with parietal infarct), 1 arteria cerebri media hemorrhage, 1 TBI (left hippocampus and pons). **CG** 11 healthy participants	**EG** mean age = 66.09; SD = 3.30 CG mean age = 66.18; SD = 2.97	The aim was a route-finding in a district of the real-world town of Graz, Austria.	Nonimmersive Virtual Reality.	Four spatial tests before and after the five VR training sessions: the Benton Test, the LPS 50+, the LVT, and the CBTT.	Route finding ability in the VR task increased over the five training sessions. Moreover, both groups improved different aspects of spatial abilities after VR training in comparison to the spatial performance before VR training.
6	Grewe et al. [52]	2014	33	**EG** 14 patients with focal epilepsy (frontal = 3, temporal = 8, central = 2, parietal = 1). **CG** 19 healthy participants	**EG** mean age = 31.29; SD = 9.44; 8 males. **CG** mean age = 31.21; SD = 14.26; 4 males	The aim was to navigate into a virtual medium-sized supermarket, modeled according to a real standard supermarket, in search of a specific list of objects.	OctaVis, semi-immersive Virtual Reality device.	BRLD-A, BRLD-B; ROCF copy, ROCF immediate and delayed recall; RWT Total Score; Digit Span Forward and Backward; VLMT immediate recallB, VLMT total learningB Trials, VLMT loss after InterferenceB, VLMT loss after delayB.	Spatial navigation and memory performance (n° of correct products, movements trajectories, time) significantly increased in the course of the 8-day training. Due to the small sample sizes in the subgroups, it could not be established the effects of different sites of epileptic foci.
7	Claessen et al. [53]	2015	6	6 stroke patients with left (N = 3), right (N = 2) and bilateral (N = 1) supratentorial stroke. No control group.	mean age = 57; SD = 8.9; 2 males	The aim was a route-finding in the Virtual Tubingen town.	Nonimmersive Virtual Reality with a joystick (Virtual Tübingen).	CBTT, TMT A-B, WAIS-III, DART, Virtual Tübingen Test (Scene recognition, Route continuation/sequence/order/progression/distance, Pointing to start/to end, Map drawing/recognition).	Navigation abilities clearly improved in one patient, partially in four cases. For other cases, were successful in adopting an alternative navigation strategy and improved on most of the trained abilities. VR was judged as highly feasible by the patients.
8	Faria et al. [54]	2016	18	**EG** 9 stroke patients. **CG** 9 stroke patients	**EG** mean age = 58 – 71; male = 44%. **CG** mean age = 53; male = 44%	The aim was to navigate in order to accomplish some common ADL’s (in a supermarket, a post office, a bank, and a pharmacy) in a virtual city with streets, sidewalks, commercial buildings, parks and moving cars.	Nonimmersive Virtual Reality with a joystick (Reh@City).	ACE, TMT A-B, Picture Arrangement Test, SIS 3.0.	VR group improved in attention, visuospatial abilities, memory, executive functions, emotion, global cognition, and overall recovery. Between comparisons showed training effect on global cognition, executive functions and attention for VR group.
9	White & Moussavi [55]	2016	1	MCI patient with probable development of AD	74 (male)	The aim was to navigate into a virtual building in search of specific targets.	Immersive Virtual Reality system with Head-mounted Display and joypad.	MoCA, VRN task (Byagowi & Moussavi, 2012), navigation diary.	The patient improved navigation during the sessions assessed with the VRN task and as reported with the wife’s diary.
10	Bate et al. [56]	2017	1	Patient with developmental prosopagnosia with concurrent topographical disorientation	58 (female)	The aim was to navigate in a virtual city, (containing six landmarks such as cinema, restaurant, pub, hotel, pharmacy, and florist) and recall the position of each landmark on a top-view map of the city.	Nonimmersive Virtual Reality with the keypad.	WAIS-III, WMS-IV, Wisconsin Card Sorting Test, CBTT, Rey’s complex figures, Picture Naming, WTAR, VOSP. Face processing tasks: CFMT, famous faces, CFPT, Ekman 60, navigational assessment: Benton, Santa Barbara Sense of direction Scale, Memory of building, ‘O clock task, route map.	Following the last session of treatment, the patient was able to form a cognitive map faster than the first one and the performance in the retrieval task was improved. A similar performance was observed at the one-week follow-up session.
11	De La Torre - Luque et al. [57]	2017	20	20 patients with a neurological diagnosis included cerebral palsy (20%), intellectual development disorder (20%) and both disorders (55%); TBI (5%).	mean age= 34.35, SD= 10.2; 13 males and 7 females.	The aim was to move through the virtual environment, and then through the equivalent real-life one and to find the same two rooms for both environments.	Semi-immersive Virtual Reality with a joystick and a mouse. A Mitsubishi® projector (model XL8U), projecting onto a × 1.5-m screen.	For the assessment of cognitive visuospatial planning and orientation, 2 tests: Porteus Maze Test; Mindscape’s Brain Trainer® 2 Maze Stair Test.	Both groups improved in a similar way, though we can say that the best. results in the virtual and the real building and generalization goals were due to virtual training. Firstly, a reduction in errors and time needed to locate the objectives in the virtual building was found after the training, so as to point out that the active navigational training showed changes. In addition, the participants had better scores in the posttest and generalization tasks in the real environment and when using maps of the building, and these tasks were not directly trained.
12	De Luca et al. [58]	2017	1	Neglect patient (subarachnoid hemorrhage, right fronto-temporal-parietal region).	57	The aim was to move in the virtual environment and manipulate specific objects, in order to realize specific associations.	Semi-immersive VR (BTs Nirvana PC System connected to a projector or a big screen).	MMSE, BIT; line crossing and bisection, letter and star cancellation, map navigation, card, and coin sorting, drawing and copying tests, phone dialing, menu and article reading, telling and setting the time.	The training enhanced spatial cognition, visual search, and attention. In addition, with standard cognitive treatment was observed a nearly complete recovery of Unilateral Spatial Neglect.
13	Serino et al. [21]	2017	28	**EG** 10 patients with AD. 8 healthy participants. **CG** 10 patients with AD	**EG** patients mean age = 86.60; SD = 6.13; 1male. healthy mean age = 86.62; SD = 6.19; 4 males. **CG** patients mean age = 88.7; SD = 3.59; 2 males	The aim was to navigate inside the virtual environment, to discover one, two or three hidden objects (i.e., a bottle of milk, a plant in a vase and a trunk) to retrieve their positions in the last phase.	Nonimmersive VR (NeuroVR software).	MMSE, Phonemic fluency, Categorical fluency, FAB, Attentional Matrices Test, Digit span test, Corsi Block-Tapping Test, Corsi Supra-Span Test.	The training enhanced spatial learning in the VR group-AD compared to control group-AD and VR healthy group improved executive functions compared to VR group-AD.
14	De Luca et al. [59]	2018	12	**EG** 6 post-stroke patients. **CG** 6 post-stroke patients	**EG**/**CG** mean age = 40; SD = 14	The aim was to move in the virtual environment and manipulate specific objects, in order to realize specific associations.	Semi-immersive VR (BTs Nirvana PC System connected to a projector or a big screen).	MoCA, FIM, FAB, AM, TMT A, TMT B, TMT A/B.	VR can be useful in potentiating the cognitive recovery in post-stroke chronic phase. It improved visuospatial and attention in the experimental group.
15	Maresca et al. [60]	2018	1	A right-handed patient affected by incomplete cervical vertebro-spinal trauma, presented with a moderate tetraparesis, mainly involving the left side.	60 (male)	The aim was to move in the virtual environment and manipulate specific objects, and to realize specific associations.	A nonimmersive virtual reality rehabilitation system (VRRS) by Khymeia, interacting with a touch screen or a magnetic tracking sensor.	MoCA, AM, TMT, digit span, RAVLI, RAVLR, Wigl’s sorting test, Raven’s colored matrices, VFT, SFT, HRS-D, HRS-A.	The combined approach using VRRS demonstrated a significant improvement in different cognitive domains as spatial abilities, executive functions, selective attention, and memory abilities.
16	Mrakic-Sposta et al. [61]	2018	10	**EG** 5 MCI patients with compromise visuospatial abilities. **CG** 5 MCI patients with compromise visuospatial abilities	**EG**/**CG** aged > 65 years; 4 males and 6 females	The aim was to navigate and to orientate inside three virtual environments (ride a bike in a park, crossroads in a city and shopping in a supermarket).	Semi-immersive scenarios with a finger touch projector and a PlayStation controller and cycle-ergometer, a Wearable smart garment (heart rate)	MMSE; RAVLT-I and RAVLT-D; ROCFT; AM; TMT-A and TMT-B; FAB; VFT.	The presented results suggest that the adopted training protocol was able to affect MMSE tasks and to increase the global cognition levels of MCI patients.

**Table 2 jcm-08-01516-t002:** Training characteristics.

	Authors	Type of Training	Single Session Duration (min)	Repetitions	Frequency/Period	Total Hours
1	Pugnetti et al. (1998)	Navigational training with active and passive conditions + recall landmarks	30	1		30 min
2	Akhutina et al. (2003)	Navigational task	30–60	6–8	within a month	3–8 h
3	Caglio et al. (2012)	Navigational training	90	15	3 times a week for 5 weeks	22.5 h
4	Grewe et al. (2013)	Navigational training + free recall of objects list and positions (at last session)	20	8	daily	2.6 h
5	Kober et al. (2013)	Navigational training + recall up to maximal three different routes	20	6	-	2 h
6	Grewe et al.	Navigational training + free recall of objects list and positions (at last session) + real-life performance	30	8	Every 1–3 days within 2 weeks	4 h
7	Claessen et al. (2015)	Navigational training	60	4	-	4 h
8	Faria et al. (2016)	Navigational training	20	12	4–6 weeks	4 h
9	White & Moussavi (2016)	Navigational training	45	21	3 times a week for 7 weeks	15.75 h
10	Bate et al. (2017)	Navigational training + recall landmarks	60-70	7	Every 3–4 days	7–8 h
11	De La Torre Luque et al. (2017)	Navigational training	20	15	daily	5 h
12	De Luca et al. (2017)	Navigational training + association of object position	45	20	5 times a week for 1 month	15 h
13	Serino et al. (2017)	Navigational training + recall object positions	30	10	3 times a week for 3–4 weeks	5 h
14	De Luca et al. (2018)	Navigational training + association of object position	45	24	3 times a week for 8 weeks	18 h
15	Maresca et al. (2018)	Navigational training	60	36	3 times a week for 12 weeks	36 h
16	Mrakic et al (2018)	Navigational training	45	18	3 times a week for 6 weeks	13.5 h

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
