# Peer review of "Neurorehabilitation of Spatial Memory Using Virtual Environments: A Systematic Review"

_jcm, 2019, doi:10.3390/jcm8101516_

Round 1

Reviewer 1 Report

Montana et al provide a detailed summary of the potential use of virtual reality for neurorehabilitation with respect to spatial memory. Methodology: an extensive literature search has been performed, albeit using only two databases. Selection criteria is non-specific- target population, choice of intervention, and outcome of interest have not been clearly reported. Spatial memory is a broad area, subcomponents of interest should be specified prior to systematic review. Evaluating risk of bias has not been included. Results: an elaborate discussion of results from the 16 included studies has been provided. The subdivision of this section appears more consistent with a narrative rather than a systematic review. Patient cohort details such as demographics, underlying pathology, sample size etc have not been reported. Discussion of duration/ intensity of VR interventions is interesting but we cannot draw firm conclusions on the basis of this review due to heterogeneity of interventions/ outcome measures/ inability to perform meta-analysis. Discussion: Again, this is of relevance as a general discussion of the use of VR in neurorehabilitation but not as a systematic review of the literature with the intended purpose of answering a specific question (as per the PICO format). Whilst Montana et al provide a narrative review of the potential of virtual reality, they do not set out a clear target question that requires a systematic review of the literature to answer. Indeed, the crux of this article is that VR has a potential role in neurorehabilitation of spatial memory, but we cannot draw any other objective conclusions at this stage. Other requirements of a systematic review such as registration with PROSPERO, or assessment of bias have not been addressed. 

Author Response

REVIEWER 1

We are very grateful to you for the time and involvement you have shown reviewing the manuscript. As you see, we have revised the manuscript and incorporated the comments and recommendations. We have highlighted the text in all the revised part in our manuscript from the original submission to facilitate the review process.

The comment we received was unpacked below in seven points to facilitate a fluid understanding of the changes made. We hope that the changes have improved the quality of the original manuscript considerably. Below we have listed your detailed comments (italics) and our responses (red).

“Methodology: an extensive literature search has been performed, albeit using only two databases. Selection criteria is non-specific- target population, choice of intervention, and outcome of interest have not been clearly reported. Spatial memory is a broad area, subcomponents of interest should be specified prior to systematic review.”

Thank you for allowing us to explain this point. We completely agree with the fact that research carried out only on two databases is a weakness of work. For this reason, the section “Limitations” has been added in which it is inserted and explained this point and others. However, we have chosen to carry out extensive research, which limited the risk of not including essential works in this research field. For this purpose, the keywords were selected, based on the PICO format, including broader words. In paragraph overmentioned, we reported these parts "This review was subject to certain limitations. Extensive literature research was performed to deliver these findings; however, only two databased were queried[...]”. You can find it on line 513.As regards the selection criteria, in the light of those meaningful comments, we proceeded to make a more precise explanation of this crucial part in our ms. At line 108 we restructured the sentence that explains the main objective of the review to better explain the rationale “According to these premises, we aimed at providing a systematic review of the experiments that purposely investigated in the field of spatial memory neurorehabilitation to comprehend if is VR navigational training, compared to treatment as usual, effective in improving navigational abilities”. Precisely because spatial memory is a large area, we have focused our attention on the specific sub-component. Thanks to your observation, we have had the opportunity to reformulate our objective better. Also, the section “Method” has been better structured, and the following paragraphs are now present: search methodology, inclusion criteria, risk of bias, and PRISMA flow diagram. In the section “inclusion criteria”, strict selection criteria have been better specified. The changes made can be read starting with line 118 until 167.

“Evaluating risk of bias has not been included.”

Thank you very much for this crucial comment; we acknowledged that we did not explain in detail the procedure. In the precedent version of the manuscript submitted we had reported, in the paragraph “PRISMA”, only “Full texts of eligible articles were retrieved and assessed by two reviewers (J.M. and C.T.) for individual selection of papers to reduce the risk of bias and resolving disagreements through consensus”. This simple sentence was not enough and made the manuscript not exhaustive and incorrect. To address this point, we add an entire paragraph entitled “Risk of bias assessment” (line 150) as explained above. Aware of the fact that it is a crucial point for any systematic review, as reported in PRISMA (D. Moher, A. Liberati, J. Tetzlaff, D. G. Altman, and P. Grp, “Preferred Reporting Items for Systematic Reviews and Meta-Analyses: The PRISMA Statement (Reprinted from Annals of Internal Medicine),” Phys. Ther., vol. 89, no. 9, pp. 873–880, 2009.) the assessment of bias was carried out according to the guidelines. In the manuscript, it is now reported as suggested in the reference handbook (J. Higgins and S. Green, Cochrane Handbook for Systematic Reviews of Interventions, Version 5. 2011):  “We follow the methods recommended by The Cochrane Collaboration Risk of Bias Tool [77] and the STROBE Statement [78]. Two reviewers (J.M. and C.T.) independently assessed the risk of bias of each included study against key criteria: random sequence generation, allocation concealment, blinding of participants, personnel and outcomes, incomplete outcome data, selective outcome reporting, and other sources of bias. The following judgments were used: low risk, high risk, or unclear (either lack of information or uncertainty over the potential for bias). Disagreements were resolved through consensus, and a third author was consulted to resolve disagreements if necessary.”

“Results: an elaborate discussion of results from the 16 included studies has been provided. The subdivision of this section appears more consistent with a narrative rather than a systematic review.” And “Discussion: Again, this is of relevance as a general discussion of the use of VR in neurorehabilitation but not as a systematic review of the literature with the intended purpose of answering a specific question (as per the PICO format). Whilst Montana et al provide a narrative review of the potential of virtual reality, they do not set out a clear target question that requires a systematic review of the literature to answer.”

It is a very good point. These are very meaningful observations for improving the readability of our ms. In Results, we better explained the VR training characteristics analyzed that could influence the treatment outcomes of navigational abilities according to the specific objectives of the review. The structure of this section has been modified to make the manuscript more readable. To facilitate a faster and more focused information search, we have added specific paragraphs for each point analyzed. Editing allowed us to make the section in line with a systematic review. We agree entirely that previously, the structure appeared more similar to a narrative revision. Thanks to your observations, we had the opportunity to improve the manuscript, hoping it could be clearer, and therefore the content could be more accessible to readers. To that end, the three sections previously present have now been subdivided with specific paragraphs for each point analyzed. You can find it in the highlighted text starting from line 206 until 435. Following the comments you provided, directly in the ms we better specified the clinical sample of each study included. We changed the structure of Table 1, to divide better the specific relevant information about the sample ( you can find a detailed explanation in response to point 5).

Moreover, we changed the form of the content in the column "VR task", in which now explain only the aim of each VR task. A second Table (Table 2) have been added to show better the VR training characteristics for each study. The division has been made with the purpose to better read out the specific characteristics of every study reported in the ms and analyzed in the referenced paragraph. We have inserted a specific section as a type of training (in which we categorized the task), number of repetition, session frequency (or period), and total hour of VR navigational training. The table has been taken up in the text for a more in-depth analysis.

 As regards the Discussion section, we started with “In the current review, we provided, although initial, positive results concerning the effect of virtual reality (VR) training on spatial memory rehabilitation, highlighting the potential of the navigational task in virtual environments (VEs) to enhance navigation and orientation abilities in patients with spatial memory disorders.” The text was elaborated to resume the different points analyzed in the results to broaden their discussion according to the question, based on PICO format,  presented at the beginning of the review (line 108 and in “method” to paragraph 2.1).

“Patient cohort details such as demographics, underlying pathology, sample size etc have not been reported.”

Thank you for this observation.  Please see the changes made according to your comment in the revised version of ms. Following your comments, we changed Table 1 with a figure better detailing the description of the sample for each study. Table 1 is now arranged and expanded into three specific columns for the reference sample (Sample (N), Sample characteristics, Mean Age (SD or range)for each study. This change was made to make this information more accessible, which previously were present, but all brought together confusingly. Thanks to the suggestion, we were able to improve access to relevant information. For the same purpose, we have also changed the column for the description of the VR task, in which currently is explained the aim of the task in virtual reality. Furthermore, the sheet on which the table is located has been rotated so as to give more space and make the content more readable. 

“Discussion of duration/ intensity of VR interventions is interesting but we cannot draw firm conclusions on the basis of this review due to heterogeneity of interventions/ outcome measures/ inability to perform meta-analysis.” And “Indeed, the crux of this article is that VR has a potential role in neurorehabilitation of spatial memory, but we cannot draw any other objective conclusions at this stage.”

We are aware of it, and we agree entirely with your observation. At the beginning of the discussions, we wrote that these are initial positive results (line 438):“In the current review, we provided, although initial, positive results concerning the effect of virtual reality (VR) training on spatial memory rehabilitation, highlighting the potential of the navigational task in virtual environments (VEs) to enhance navigation and orientation abilities in patients with spatial memory disorders.” We, therefore, hope that the literature could soon be enriched with proper clinical trials to investigate the effectiveness of  VR navigational training in more specific clinical populations. For this reason, we have also included this part in the limitations section. Also in the discussions, on line 497 we refer to this “The difficulty in finding homogeneous groups of patients is the reason why most of the experimental studies reported in the literature are based on the treatment of a single patient or a tiny group of patients […]”. So at this stage, we do not yet have consistent outcome measures, but an initial analysis may still be relevant to guide future studies. 

Thank you for your valuable contribution to our work; we feel that our manuscript is much more improved, further to your reviews, especially in its clarity for readers.

Reviewer 2 Report

Neurorehabilitation of spatial memory using virtual 3 environments: a systematic review

Journal of Clinical Medicine

The study is a systematic review of the literature on using virtual environments as a neurorehabilitation platform for spatial memory. The authors conduct a nice review the literature and offer recommendations based on the findings.

Only minor edits are noted for example using was in place of has been and were in place of have been:

Abstract

Line 16 change “has been carried out” to “was carried out…”

Results

Line 149 change “have been analyzed” to “was analyzed” to show it is done.

Results

Table 1,first study under training is “A” session indicate one session was completed? If so, I would replace a with To simplify Table 1, may be easier if age ranges were used to describe the sample for each study as each is variable and more difficult to follow. In general, try to be more consistent in information being reported across studies, so it is easier to compare. Hard to Follow tables in portrait view. Heading 3.3 Which assessment has been used. Which are the outcomes is a little akward as written. Maybe replace “which” with “

Conclusion

Line 417 …VR systems are the best ecological tool…. Seems a bit overstated.

Overall, the study is a useful contribution to the literature and only needs minor edits.

Author Response

REVIEWER 2

We are very grateful to you for the time and involvement you have shown reviewing the manuscript. As you see, we have revised the manuscript and incorporated the comments and recommendations. We have highlighted the text in all the revised part in our manuscript from the original submission to facilitate the review process.

The comment we received was unpacked below in five points to facilitate a fluid understanding of the changes made. We hope that the changes have improved the quality of the original manuscript considerably. Below we have listed your detailed comments (italics) and our responses (red).

Line 16 change “has beencarried out” to “was carried out…” And Line 149 change “have beenanalyzed” to “was analyzed” to show it is done.

Thank you for this critical suggestion for improving the readability and correctness of English of our ms.  Please see the changes made according to your comment in the revised version of ms.

“Results Table 1, first study under training is "A”session indicate one session was completed? If so, I would replace a with To simplify Table 1, may be easier if age ranges were used to describe the sample for each study as each is variable and more difficult to follow. In general, try to be more consistent in information being reported across studies, so it is easier to compare. Hard to Follow tables in portrait view.”

Thank you for this observation. It is an exciting way to better organizing our ms. Following your comments, we changed Table 1 with a figure better detailing the description of the sample for each study. We have added two columns to the table (in addition to Sample (N): Sample characteristics, Mean Age - SD or range), to make information more transparent and more comfortable to identify.  For the same purpose, we have also changed the column for the description of the VR task, in which currently is explained the aim of the task in virtual reality. Furthermore, the sheet on which the table is located has been rotated so as to give more space and make the content more readable.

A second Table (Table 2) have been added to show better the VR training characteristics for each study. We have inserted a specific section as a type of training (in which we categorized the task), number of repetition, session frequency (or period), and total hour of VR navigational training.

“Heading 3.3 Which assessment has been used. Which are the outcomes is a little awkward as written.”

Thank you for your significant suggestions for improving the readability of our ms. As suggested, we proceeded to modify the title to make the content clearer. We have replaced with "3.3. WHICH ASSESSMENT AND OUTCOMES FOR SPATIAL MEMORY?" and we have divided the text into two further sub-sections. The division of the paragraphs was done with the intention of better guiding the reading of the manuscript, leading the reader to identify the sections of interest quickly.

“Conclusion Line 417 …VR systems are the bestecological tool…. Seems a bit overstated.”

Thank you for allowing us to improve the fluency in this part. We have replaced the form of the sentence in "In this view, VR systems offer impressive opportunities as an ecological tool which is currently available for neuropsychologists to assess…..”. Thank you for your contribution to our work, we feel that thanks to this we have improved its value. We much appreciate that you found interesting in our manuscript.

Round 2

Reviewer 1 Report

I am satisfied with the current manuscript and the amendments the authors have made.